# Genome-Wide Association Study of Agronomic and Physiological Traits Related to Drought Tolerance in Potato

**DOI:** 10.3390/plants12040734

**Published:** 2023-02-07

**Authors:** Alba Alvarez-Morezuelas, Leire Barandalla, Enrique Ritter, Jose Ignacio Ruiz de Galarreta

**Affiliations:** NEIKER-Basque Institute for Agricultural Research and Development, Basque Research and Technology Alliance (BRTA), Campus Agroalimentario de Arkaute, 01192 Arkaute, Spain

**Keywords:** *Solanum tuberosum*, drought stress, GWASpoly, SNP

## Abstract

Potato (*Solanum tuberosum* L.) is often considered a water-sensitive crop and its production can be threatened by drought events, making water stress tolerance a trait of increasing interest. In this study, a panel of 144 tetraploid potato genotypes was evaluated for two consecutive years (2019 and 2020) to observe the variation of several physiological traits such as chlorophyll content and fluorescence, stomatal conductance, NDVI, and leaf area and circumference. In addition, agronomic parameters such as yield, tuber fresh weight, tuber number, starch content, dry matter and reducing sugars were determined. GGP V3 Potato array was used to genotype the population, obtaining a total of 18,259 high-quality SNP markers. Marker-trait association was performed using GWASpoly package in R software and Q + K linear mixed models were considered. This approach allowed us to identify eighteen SNP markers significantly associated with the studied traits in both treatments and years, which were related to genes with known functions. Markers related to chlorophyll content and number of tubers under control and stress conditions, and related to stomatal conductance, NDVI, yield and reducing sugar content under water stress, were identified. Although these markers were distributed throughout the genome, the SNPs associated with the traits under control conditions were found mainly on chromosome 11, while under stress conditions they were detected on chromosome 4. These results contribute to the knowledge of the mechanisms of potato tolerance to water stress and are useful for future marker-assisted selection programs.

## 1. Introduction

Climate change is causing negative effects on crop production, both through biotic stresses and abiotic stresses such as temperature stress, drought and salinity [1]. The impact of climate change on crop yield and quality will vary depending on the area and crop system [2]. In Spain, crops are mostly grown using artificial irrigation systems, which optimize the limited water available. However, the availability of water resources has been decreasing in recent years and in the future, it will be necessary to increase the amount of irrigation or even to irrigate in rainfed areas. Therefore, it will be essential to cultivate more water-efficient materials [3].

Potato (*Solanum tuberosum* L.) is one of the most important crops in the world with an annual production of 359 million tons of tubers (FAOSTAT, 2020). It is a highly-valued crop as it can grow in a wide range of environments, is very versatile in terms of uses, is a short-duration crop, and 85% of its biomass is edible [4,5]. Potatoes are relatively water-efficient and compared to other crops, produce more calories per unit of water used [6,7]. However, this crop also has a high irrigation requirement and it is considered a drought-sensitive crop. The drought susceptibility of potatoes is associated with their shallow and sparse root system, but canopy development and variety also play an important role in water stress tolerance [4,8]. Drought is one of the main factors limiting yield, particularly in susceptible crops such as potatoes. If potato crops are not adapted to water stress, a loss ofyield between 18% and 32% is estimated for the year 2050, although with adaptation there would be a reduction of between 9 and 18% for the period 2040–2069 [9]. It is difficult to estimate the global yield loss due to water stress alone, as other abiotic stresses such as temperature, solar radiation or salinity are closely related. However, some studies have reported a decrease between 15% and 91% in potato yield under water stress conditions [10,11,12].

Potato breeding activities in recent years have focused on searching for regions of the genome related to tuber quality traits [13,14], agronomically-important traits [15], floral traits [16], root and stolon traits [17], and nitrogen use efficiency [18], and markers have also been developed for applying in marker-assisted selection for resistance to some diseases such as common scab [19,20] or *Phytophthora infestans* [21,22].

Drought tolerance is a complex trait which depends on several factors such as the duration of the stress, the severity of the drought, and the developmental stage of the plant. Stress in the early growth stage is considered the most harmful [23,24]. From a physiological point of view, survival or recovery is the major objective in plant stress tolerance, but from an agricultural point of view, crop yield is the trait that determines crop drought tolerance [25]. Yield decrease is mainly associated with inhibition of photosynthesis, decrease in stomatal conductance to prevent water loss through transpiration, and reduction of leaf area [11,24,26,27].

Breeding for drought tolerance is challenging and absolutely essential under the expected climatic changes that could lead to more frequent periods of low water supply. Genetic basis of drought tolerance is complex, but there are tools such as the DroughtDB database which collects genes of interest for drought stress in plants and helps us understand the mechanisms of tolerance [28]. Although an enormous amount of knowledge has been gained about drought tolerance in recent years, we are still far from understanding all the underlying mechanisms and signaling pathways involved [25]. Water stress tolerance traits are polygenic and affected by several minor alleles. Therefore, a deeper understanding of the loci and alleles involved is needed.

Traditional potato breeding has certain difficulties due to the heterozygous nature of tetraploid potatoes, and furthermore, allelic combinations and genetic effects become even more complex when dealing with quantitative polygenic traits such as water stress tolerance [29]. The collection of accurate phenotypic data for the traits of interest in the study population is a major challenge, as these assays should be multi-year and multi-environment and should have a sufficient number of genotypes population under study [30].

The potato genome is comprised of 12 chromosomes and has an average size of approximately 840 Mbp. For a few years now, the complete genome sequence is available and allowed the development of Single Nucleotide Polymorphisms (SNP) arrays by the potato community [31]. Several generations of SNP arrays were generated, building on the original Infinium 8303 SNP array [32]. In recent years, advances in sequencing have been developed, sequencing costs have decreased, and the number of reads has increased [33]. 

Association mapping, also known as linkage disequilibrium (LD) mapping, is a powerful tool for the association of a phenotype with a genotype and the identification of causal genes/loci [34]. One of the most attractive aspects of association mapping is that it is not necessary to establish mapping families, and instead historical recombination events can be explored at the population level [35,36]. The absence of biparental crosses for identifying QTL makes association mapping easier and less expensive [37].

In this study, we have performed Genome Wide Association Studies (GWAS) with the aim of identifying QTLs associated with physiological and agronomic traits of interest for potato breeding under water stress and unstressed conditions, in order to accelerate the selection processes in potato breeding programs.

## 2. Results

### 2.1. Phenotypic Data Analysis

Analysis of variance (ANOVA) showed highly significant differences for all traits between genotypes, between treatments and interactions between genotypes and treatments (G × T) in both years (Table 1). Descriptive statistics for the traits are provided in the Appendix A.

The correlation of physiological and yield-related variables between control and stressed samples was studied (Figure 1). Yield is one of the most important traits when looking for tolerance to abiotic stresses. We saw that the yields under control conditions and under water stress conditions were correlated with more or less the same traits, especially with number and weight of tubers under both control and drought conditions. Yield_C and Yield_D was also correlated with most of the physiological parameters and the highest correlations occurred 70 days after planting (DAP). All correlations were positive, except for FLUOR 50, FLUOR 70, dry matter and starch.

### 2.2. Population Structure Analysis and Linkage Disequilibrium 

STRUCTURE software revealed that the study population was formed by two subpopulations of 133 and 11 genotypes respectively, since the obtained delta K value was 2 (Appendix A). The probabilistic assignment of each genotype to belong to one of the assigned groups was also performed for deriving the corresponding values of the Q matrix (Appendix A). These results indicate that there was genetic diversity in the population with different structural dimensions, which was also considered for the association analysis. A genetic distance matrix was performed between all genotypes to evaluate the genetic diversity and it was observed that the highest value between two varieties was 0.4, the minimum value was 0.26 and the mean value was 0.37.

Linkage disequilibrium (LD) decay was determined using the filtered SNP data. In our study, the genetic distance between markers was calculated as the point of intersection between the half decay r^2^ value of the genome and the smoothing spline regression model fitted to LD decay (Appendix A).

### 2.3. Genome-Wide Association Analysis

The total of 31,190 markers were filtered to ensure the quality of the SNPs, removing markers with a missing value rate higher than 10% and those with a minor allele frequency below 0.05, obtaining 18,259 SNP markers. These SNP markers provide a genome-wide coverage along the 12 chromosomes of tetraploid potatoes (Table 2).

The association mapping was performed with kinship correction to minimize false positive associations. The Q + K model was used with the 18,259 high-quality SNP markers and the panel of 144 accessions. The results of the Q-Q plots indicate that the observed −log_10_(P) values are in accordance with the expected −log_10_(P) values (Appendix A).

The results of the association analysis are presented as marker-trait associations to get an overall impression of the effect of water stress in our population. In this study, eighteen QTLs were identified above the Bonferroni threshold. Five of these QTLs were associated with two of the traits measured under control conditions, while the rest were associated with traits measured in plants under water stress.

Two SNP markers associated with chlorophyll content measured at 70 DAP were found, one on chromosome 6 (PotVar0039950) and the other on chromosome 11 (solcap_snp_c2_15287). Two other QTLs were also found on chromosome 11, which in this case were associated with the number of tubers in control plants (solcap_snp_c2_37217 and ST4.03ch11_2070850). The marker solcap_snp_c2_15676, located on chromosome 5, was also associated with this trait (Table 3, Figure 2).

If we observe the physiological parameters under drought conditions we can see that most of the associations occurred with measurements taken at 70 DAP. Two QTLs associated with Normalized Difference Vegetation Index (NDVI) were found, both on chromosome 4 (solcap_snp_c2_43735 and PotVar0113919). The marker solcap_snp_c2_45637 on chromosome 1 was also found to be associated with stomatal conductance and the marker PotVar0039950 on chromosome 6 was associated with leaf chlorophyll content. Although almost all associations were found in measurements taken at 70 DAP, the NDVI was also associated with one marker (solcap_snp_c1_6462) in the first stress phase, at 50 DAP (Table 3, Figure 2).

One of the most important parameters when assessing stress tolerance is the maintenance of crop yield. In this case we saw that the marker solcap_snp_c2_26653, located on chromosome 8, was associated with yield under water stress conditions, and that it is co-localized with the osmotin gene (Soltu.DM.08G027260.1). The markers PotVar0064470 and solcap_snp_c2_55085, located on chromosomes 10 and 11 respectively, were associated with tuber number under stress conditions. The trait for which the most associated QTLs were found was the content of reducing sugars under drought conditions. One of them (solcap_snp_c1_3746) was found on chromosome 2, while the other four were located on chromosome 4, and three of them (solcap_snp_c2_55785, solcap_snp_c2_55783, solcap_snp_c2_55775) co-localized with the same gene, leucine-rich receptor-like protein kinase family protein (Table 3, Figure 2).

## 3. Discussion

Thanks to new massive sequencing techniques and the development of chips such as the GGP Potato 35K array used in this study, we can obtain a global view of the genome and select regions and genes of interest related to the desired trait [32,38]. The traits evaluated in this work have complex inheritance patterns that make the task of existing mapping technologies to detect the underlying genetics even more difficult. Different studies have analysed the heritability of yield and its components under control and water stress conditions. These traits under control conditions have a fairly acceptable heritability of around 0.7 [39], but it is not very clear how water stress affects the heritability of these traits. In some studies, it drops to 0.06 [40], while in other studies this decrease was much lower [41].

When analysing multiple tests one must address the problem of false positives, so it is important to adjust the *p*-value of each marker when performing the statistical analysis [42]. In our study, we can observe that the FDR values are higher than the Bonferroni *p*-values. The Bonferroni correction is the most commonly used in association studies, but this method is very strict and can sometimes fail to identify important associations, so the FDR correction is usually used [43,44]. 

In this study, QTLs related to chlorophyll content measured at 70 DAP and tuber number under control conditions were identified. These two parameters also showed a significant positive correlation, indicating that the amount of chlorophyll in the leaves of the plants has an effect on the number of tubers. 

When plants are under water stress, one of the tolerance mechanisms is the inhibition of photosynthesis, and as a consequence, chlorophyll content decreases. Chlorophyll content was significantly associated with two SNPs, solcap_snp_c2_15287 and PotVar0039950. The solcap_snp_c2_15287 (Soltu.DM.11G023130) was co-localized with a gene encoding for a “P-loop containing nucleoside triphosphate hydrolases superfamily protein”, which is a type of hydrolase that catalyses the hydrolysis of the beta-gamma phosphate bond of a bound nucleoside triphosphate (NTP), and the obtained energy from this reaction is used to make conformational changes in other molecules [45]. In an assay on water-stressed *Arabidopsis*, they found an association between two P-loop-containing nucleoside triphosphate genes and proline content, which is closely related to plant response to drought [46]. Another study in rice showed that a new DEAD-box helicase ATP-binding protein (OsABP), a kind of P-loop containing nucleoside triphosphate hydrolase, was upregulated in response to multiple abiotic stresses, including NaCl, dehydration, ABA, and blue and red light [47]. 

The PotVar0039950 marker was found to be associated with the SPAD70 trait under both control and water stress conditions. This marker co-localizes with a “Radical SAM superfamily protein” gene (Soltu.DM.06G028800.1) and is located on chromosome 6. Radical SAM is a designation for a superfamily of enzymes that are involved in numerous processes, such as enzyme activation, post-transcriptional and post-translational modifications, lipid metabolism, or biosynthesis of antibiotics and natural products [48]. In a previous study in *Sonneratia apetala* they found that the SAMS1 gene was related to this group of proteins and indicated that SAMS1 enhanced the plant’s cold resistance by enhancing the biosynthesis of S-adenosyl-L-methionine (SAM). In addition, SAMS1 is also involved in ethylene biosynthesis, which is closely related to the plant’s response to drought stress [49].

Normalized Difference Vegetation Index was associated with PotVar0113919 marker, which co-localized with the ascorbate peroxidase gene (Soltu.DM.04G030200.1). Ascorbate peroxidase (APX) is an enzyme essential for protecting chloroplasts and other parts of the cell from damage caused by reactive oxygen species, and its production increases when plants are exposed to unfavorable environmental conditions [50]. The expression of APX encoding genes is modulated by those environmental stimuli, such as drought [51]. Other studies in cowpea and wheat showed in sensitive cultivars an increase in APX transcripts in response to water stress [52,53]. Likewise in potato, an increase in ascorbate peroxidase activity was observed under drought and heat stress treatments in three of the four tested varieties [54]. In our study, this gene is associated with NDVI70 under stress conditions, similar to another study where ascorbate peroxidase concentrations were correlated with photosynthesis, Fv/Fm and chlorophyll parameters [55].

The increase in the yield under water deficit was associated with solcap_snp_c2_26653 on chromosome 8 and is co-localized with the osmotin gene (Soltu.DM.08G027260.1). Osmotin is a multifunctional protein. Its overexpression induces abiotic stress tolerance, lowering the osmotic potential under stress [56]. Studies in cotton and tomato showed that the overexpression of the osmotin gene had a protective role and enhances drought stress tolerance [57,58]. Increases in leaf expansion, chlorophyll and relative water content were observed due to overexpression of osmotin in transgenic sesame plants and were fully recovered after rewatering [59].

Tuber numbers under stress conditions were associated with two SNP markers, PotVar0064470 and solcap_snp_c2_55085. PotVar0064470 was co-localized with an “Alternative oxidase family protein” gene (Soltu.DM.11G001000.2). Alternative oxidase (AOX) activity is important for maintaining photosynthetic electron transport under stress, and also helps plants cope with excess energy under drought, by avoiding the over-reduction of chloroplast electron carriers [60,61]. During severe or prolonged mild drought stress in *Nicotiana tabacum*, the amount of AOX protein was important for maintaining the photosynthetic rate and improving growth during prolonged water deficit [62]. In our study, the number of tubers was significantly correlated with photosynthesis-related parameters such as chlorophyll content or NDVI, which confirms the protective function of AOX. Also associated with tuber number was solcap_snp_c2_55085, which co-localized with “Transketolase” gene. Transketolase (TK) is an enzyme that participates in both the pentose phosphate pathway in all organisms and the Calvin cycle of photosynthesis [63]. In a study with wheat plants, the decrease in transketolase level suggested the suppression of the two pathways in the leaves of drought-stressed plants [64]. However, in studies with transgenic rice, co-overproduction of Rubisco and transketolase did not improve photosynthesis [65].

The content of reducing sugars under drought conditions was associated with five QTLs. One of these QTLs was solcap_snp_c2_25284 and co-localized with sucrose transporter (Soltu.DM.04G031670.1). Cellular accumulation of soluble sugars during drought stress influences the expression of sugar transporters [66], which is in agreement with the results obtained in our study. In potato, some studies have also analyzed the export of sucrose from the source to the leaves by analyzing the expression of genes related to sucrose transporters (SWEETs and SUTs), which are involved in stress response [24,67]. The markers solcap_snp_c2_55785, solcap_snp_c2_55783, and solcap_snp_c2_55775 were on chromosome 4 and are co-localized with the same gene, leucine-rich receptor-like protein kinase family protein (Soltu.DM.04G031690.1). Studies in rice showed that overexpression of LRK, which encodes a leucine-rich receptor-like kinase, increased drought tolerance [68,69]. A potato gene, StLRPK1, encoding a protein belonging to leucine-rich repeat receptor-like kinases was identified, and the results suggest that StLRPK1 may participate in the responses against environmental stresses in potato, which is in accordance with our results [70]. 

In this study we found markers associated with the evaluated physiological traits. Other authors have previously reported QTLs and genomic regions associated with chlorophyll content, chlorophyll fluorescence and NDVI in water stress assays in other populations, indicating that these results are robust [71,72,73]. For yield-related parameters, we found markers related to yield, tuber number and reducing sugar content as in previous studies reporting QTLs associated with these traits [74,75,76].

One additional, important aspect to consider is the validation of the significant SNP markers by expression analyses in control and water stress conditions using RT-qPCR in more sensitive and more tolerant genotypes. This aspect will be considered in a follow-up publication. 

## 4. Materials and Methods

### 4.1. Plant Material and Location

A total of 144 tetraploid potato genotypes belonging to *Solanum tuberosum* ssp. *tuberosum* were used in this study, representing a wide range of parents used in breeding programs. The field experiments were performed in the facilities of NEIKER research center in Spain (42°51′05.7″ N, 2°37′13.2″ W) during the years 2019 and 2020.

### 4.2. Experimental Design

The trials were conducted from May to September in both years, and the climatic conditions at the experimental field in terms of average maximum and minimum temperature; humidity and total precipitation are shown in Table 4. The experimental design in each year included two blocks, irrigated (control) and non-irrigated treatments. In each block the genotypes were planted in a completely randomized experimental design with two replicates of five plants each, at a distance of 0.30 m between plants and 0.75 m between rows. The irrigation strategy in the case of the irrigated control field was based on the weekly replenishment of the accumulated water deficit from the third week of June onwards. For the estimation of the doses of each of the irrigations, a daily soil water balance was calculated using the FAO56 dual coefficient model [77] and the meteorological data recorded by the Arkaute weather station belonging to the EUSKALMET network.

### 4.3. Phenotypic Data Collection

Four physiological traits were measured in each genotype at two different dates, 50 days after planting (DAP) and 70 DAP. The chlorophyll content (CC) was measured using a SPAD-502 chlorophyll meter (Konica Minolta, Osaka, Japan) in the last fully-expanded leaf in three plants of each replicate and treatment. Photochemical efficiency of PSII was measured in leaves exposed to light (Fv’/Fm’) using a fluorimeter (FluorPen FP 100, Photon Systems Instruments, Drasov, Czech Republic), likewise in the last fully-expanded leaf in three plants of each replicate and treatment. Stomatal conductance (gs, mmol H_2_O m^−2^ s^−1^) was measured using a porometer (Leaf Porometer, Decagon Devices, Pullman, Washington, EEUU) in the last fully-expanded leaf in one plant of each replicate and treatment. Normalized difference vegetation index (NDVI) was measured in each replicate using a Rapidscan (RapidScan CS-45, Holland Scientific, Lincoln, EEUU). Plants were scanned from 0.5 m above the crop canopy in five plants of each replicate along the row direction. Three leaves per replicate and variety were collected at 70 DAP from each of the treatments to estimate leaf area and leaf circumference values using ImageJ software v.1.8.0. 

Plants were harvested at 127 DAP in 2019 and 126 DAP in 2020 to allow late cultivars to complete their cycle. The whole experiment was harvested at once and total yield, tuber number per plant and average tuber fresh weight was assessed in each replicate. The five plants from each replicate were harvested together and the total value was divided by five to get yield and tuber number for each plant. Tuber weight was calculated as yield/tuber number. Dry matter content was measured in two replicates of each variety and treatment. Tubers were weighed immediately after harvest (FW). After 72 h at 80 °C, they were weighed again to obtain the dry weight (DW). The starch content was calculated with the following formula [78]: $$\mathrm{Starch}=\left(\frac{\mathrm{DW}}{\mathrm{FW}}*\;100-6.0313\right)*10$$

The determination of reducing sugars content present in the samples was assessed by spectrophotometry based on the reduction of dinitrosalicylic acid [79]. Two replicates per variety and treatment were analyzed. The potatoes were peeled and mashed into a homogeneous juice. A total of 0.3 g of the mixture was weighed and 1 ml of distilled water and 2 mL of dinitrosalicylic acid were added. Then the samples were heated at 100 °C in a water bath with stirring for 10 min. Afterwards, the samples were diluted with distilled water and the absorbance was measured in the UV-VIS spectrophotometer at 546 nm. The percentage of reducing sugars was calculated as follows: %reducing sugars=(absorbance−0.00385)∗1.07893

Analysis of variance (ANOVA) was performed on the data of both years for each parameter using Rstudio v.2022.12.0 (R Core Team, 2017) and the mean values of all traits were used to calculate the marker-trait associations.

### 4.4. DNA Extraction and Genotyping

Genomic DNA was extracted from 144 fresh potato leaves using innuPREP Plant DNA Kit (Analytik Jena, Jena, Germany) following the manufacturer’s instructions. DNA concentration and quality were measured using a NanoDrop 2000 spectrophotometer (Thermo Fisher Scientific, Waltman, MA, USA). The extracted DNA was sent to Neogen (Scotland, UK) for genotyping with the GGPv3 Potato 35K array. The software Genome Studio (Illumina, San Diego, CA, USA) was used for genotype calling, scoring four alleles per locus. The total set of markers obtained was filtered to ensure the quality of the SNPs, removing markers with a missing value rate higher than 10% and those with a minor allele frequency below 0.05.

### 4.5. Population Structure, Linkage Disequilibrium and GWAS Study

The population structure matrix (Q-matrix) was analyzed using K-values ranging from 1 to 10 for the entire population with 18,259 SNP markers with Structure v.2.4 software [80]. Three independent analyses were performed for each K-value. In this analysis, the length of the burn-in period was 100,000, with 100,000 MCMC replications after burn-in. The optimal value of K was identified using a previously-developed method based on delta K (∆K) [81] in the Structure Harvester website [82]. The relationship between genotypes and the genetic diversity in the population was calculated from the SNP marker data using TASSEL software [83].

Linkage Disequilibrium was estimated for high-quality SNPs after filtering using TASSEL software [83]. The pairwise squared allele-frequency correlations (r^2^) between SNP markers were calculated with a sliding window of 50 SNPs. These results were plotted against physical distance and an internal trend line was drawn as a non-linear logarithmic regression curve to estimate LD decay using R [84].

Association mapping analysis was performed with the phenotype and genotype data using the statistical package GwasPoly [85] developed for R software (R Core Team, 2017). The mixed model was used to perform association analysis with correction for kinship (K) and for sub-populations (Q). To correct for multiple testing, we used the 5% Bonferroni threshold (−log_10_(P) = 5.01).

## Figures and Tables

**Figure 1 plants-12-00734-f001:**
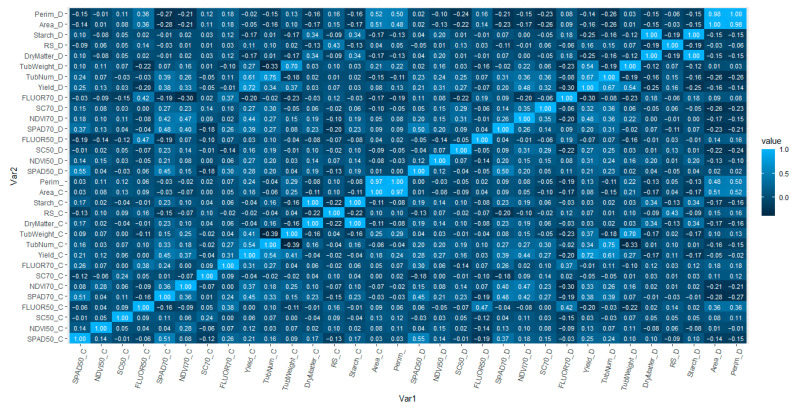
Pearson’s correlations between different physiological and agronomical traits under control (indicated with postfix “_C”) and water stress conditions (indicated with postfix “_D”) in a panel of 144 potato varieties.

**Figure 2 plants-12-00734-f002:**
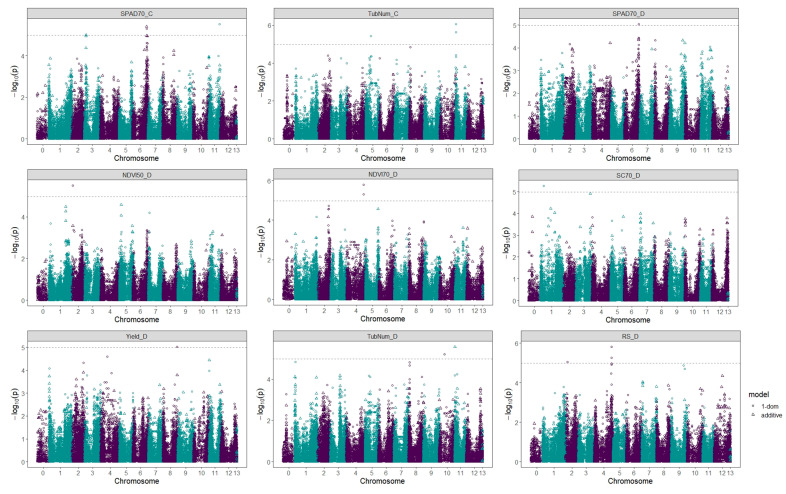
Manhattan plots for the traits with significant SNPs associated under control and drought stress conditions in 144 potato varieties.

**Table 1 plants-12-00734-t001:** Analysis of variance between genotypes (G) and treatments (T) in 144 tetraploid potato varieties.

Trait	F Value					
	2019			2020		
	Genotype(G)	Treatment(T)	G × T	Genotype(G)	Treatment(T)	G × T
SPAD_50	4.83 ***	15.17 ***	1.39 **	8.57 ***	5.51 *	1.55 ***
NDVI_50	2.53 ***	411.95 ***	2.37 ***	2.26 ***	38.25 ***	1.52 **
SC_50	3.89 ***	293.64 ***	3.95 ***	6.61 ***	275.80 ***	2.18 ***
FLUOR_50	7.72 ***	148.46 ***	2.79 ***	3.81 ***	3.81 ns	3.56 ***
SPAD_70	6.58 ***	62.22 ***	2.20 ***	8.63 ***	7.04 **	1.56 ***
NDVI_70	4.30 ***	360.85 ***	2.58 ***	2.89 ***	61.99 ***	1.75 ***
SC_70	4.49 ***	322.74 ***	3.76 ***	4.23 ***	140.85 ***	1.72 ***
FLUOR_70	6.21 ***	584.25 ***	3.10 ***	3.72 ***	252.39 ***	1.57 ***
Yield	19.63 ***	900.35 ***	5.20 ***	7.88 ***	627.35 ***	1.89 ***
TubNum	11.63 ***	352.54 ***	1.98 ***	6.25 ***	17.57 ***	1.54 ***
TubWeight	8.97 ***	859.01 ***	2.62 ***	6.24 ***	541.72 ***	1.95 ***
DryMatter	1453.69 ***	1200 ***	802.88 ***	1326.9 ***	1200 ***	530 ***
RS	51.10 ***	1200 ***	30.74 ***	32.97 ***	72.53 ***	18.85 ***
Starch	1497.04 ***	1200 ***	853.63 ***	1388.46 ***	1200 ***	558.55 ***
Area	5.58 ***	358.25 ***	2.56 ***	3.58 ***	178.29 ***	1.58 ***
Perim	6.24 ***	256.34 ***	2.14 ***	3.69 ***	189.32 ***	1.67 ***

*, **, *** Significant at *p* = 0.05, *p* = 0.01 and *p* = 0.001, respectively.

**Table 2 plants-12-00734-t002:** Number of SNPs per chromosome before and after filtering and size of each chromosome. CH01 to CH12 refers to each of the 12 potato chromosomes, CH00 are control markers that are not associated with any chromosome and CH13 refers to the chloroplast.

Chromosome	Number of SNPs (Total)	Number of SNPs (Filtered)	Chromosome Length (bps)
CH00	464	156	
CH01	3958	2486	88,663,952
CH02	3335	1914	48,614,681
CH03	2919	1637	62,190,286
CH04	2798	1611	72,208,621
CH05	2538	1520	52,070,158
CH06	2390	1461	59,532,096
CH07	2457	1407	56,760,843
CH08	2043	1234	56,938,457
CH09	2204	1296	61,540,751
CH10	1865	1061	59,756,223
CH11	2249	1361	45,475,667
CH12	1942	1098	61,165,649
CH13	28	17	155,312
Total	31,190	18,259	810,654,046

**Table 3 plants-12-00734-t003:** Significant SNPs associated with evaluated physiological and agronomical traits under control and drought stress conditions in 144 potato varieties.

Trait	Marker	Chrom.	Position	Ref	Alt	Effect	R^2^	*p*-Value	FDR	Biological Function
SPAD70_C	PotVar0039950	6	53985614	C	T	−2.07	0.0183	2.14 × 10^−2^	0.0361	Radical SAM superfamily protein
SPAD70_C	solcap_snp_c2_15287	11	41743380	A	G	−4.20	0.0681	7.05 × 10^−6^	0.0138	P-loop containing nucleoside triphosphate hydrolases superfamily protein
TubNum_C	solcap_snp_c2_15676	5	18718517	G	T	−25.38	0.0438	0.0003	0.0222	RNA-binding CRS1/YhbY (CRM) domain-containing protein
TubNum_C	solcap_snp_c2_37217	11	1818959	A	G	−32.80	0.0006	0.0486	0.05	-
TubNum_C	ST4.03ch11_2070850	11	2070850	A	T	42.25	0.0153	0.0355	0.0416	Di-glucose binding protein with Kinesin motor domain
NDVI50_D	solcap_snp_c1_6462	2	2450782	G	T	0.03	0.0744	2.53 × 10^−6^	0.0027	Plant protein with unknown function
SPAD70_D	PotVar0039950	6	53985614	C	T	−3.65	0.0712	4.25 × 10^−6^	0.0083	Radical SAM superfamily protein
NDVI70_D	solcap_snp_c2_43735	4	64055406	A	G	−0.07	0.0095	0.0432	0.0444	GroES-like zinc-binding dehydrogenase family protein
NDVI70_D	PotVar0113919	4	64089292	A	G	−0.07	0.0049	0.0461	0.0472	Ascorbate peroxidase
SC70_D	solcap_snp_c2_45637	1	12022163	A	G	−182.03	0.0729	3.26 × 10^−6^	0.0055	Hypothetical protein
Yield_D	solcap_snp_c2_26653	8	54286889	G	T	−0.75	0.071	4.37 × 10^−6^	0.0111	Osmotin
TubNum_D	PotVar0064470	11	787325	G	T	−10.25	0.0614	2.03 × 10^−5^	0.0166	Alternative oxidase family protein
TubNum_D	solcap_snp_c2_55085	10	20334943	A	G	27.36	0.0586	3.23 × 10^−5^	0.0194	Transketolase
RS_D	solcap_snp_c1_3746	2	7050595	C	T	0.14	0.034	0.0016	0.025	Cofactor assembly of complex C
RS_D	solcap_snp_c2_25284	4	65872176	A	G	0.10	0.0041	0.0277	0.0388	Sucrose transporter
RS_D	solcap_snp_c2_55785	4	65970953	G	T	0.11	0.0952	0.01	0.0277	Leucine-rich receptor-like protein kinase family protein
RS_D	solcap_snp_c2_55783	4	65971150	A	G	0.11	0.0952	0.01	0.0305	Leucine-rich receptor-like protein kinase family protein
RS_D	solcap_snp_c2_55775	4	65972399	C	T	0.11	0.0952	0.01	0.0333	Leucine-rich receptor-like protein kinase family protein

**Table 4 plants-12-00734-t004:** Maximum and minimum temperatures, humidity and precipitation in the experimental field for years 2019 and 2020.

**Year 2019**					
	**14–31 May**	**June**	**July**	**August**	**1–17 September**
Max. temperature (°C)	18.1	25.4	7.1	27.5	22.2
Min. temperature (°C)	5.7	10.0	3.1	13.0	10.0
Humidity (%)	79.9	70.3	2.3	73.5	75.1
Precipitation (L/m^2^)	33.9	17	2.1	24.7	30.5
**Year 2020**					
	**26–31 May**	**June**	**July**	**August**	**1–28 September**
Max. temperature (°C)	26.8	22.2	6.4	27.7	24.6
Min. temperature (°C)	9.2	0.7	2.4	13.0	11.0
Humidity (%)	68.3	7.2	3.2	72.3	71
Precipitation (L/m^2^)	0	5.8	8.8	31.4	33.5

## Data Availability

Not applicable.

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
