# Peer review of "Genome-Wide Association Study of Agronomic and Physiological Traits Related to Drought Tolerance in Potato"

_plants, 2023, doi:10.3390/plants12040734_

Round 1

Reviewer 1 Report

Dear Authors,

Just minor corrections:

Line 19. Tuber number – change to number of tubers?

Figure 1. Which traits marked “D” and which marked ЭСЭ, where are data related to water controlled and water stress conditions?

Suppl 2 – What are 0 -13 plots for each trait? Explain.

Line 110. 70 DAP – should be : 70 days after planting (DAP).

112. It is better to show in supplement STRUCTURE results for K=2, 3, 4 etc to make sure K=2 is a correct value.

Line 200. Reactive oxygen species are not mentioned anywhere more in the text. So, acronym (ROS) is not necessary.

Thank you,

Reviewer 2 Report

See my comments in attached pdf

Reviewer 3 Report

The manuscript "Genome-wide association study of agronomic and physiological traits related to drought tolerance in potato" submited by Alvarez-Morezuelas carried out genome- wide association study on drought tolerance of potato and screen drought candidate genes, which is of great significance for potato resistance breeding. However, it will be more perfect if the author can analyze the tissue expression and drought induced expression of candidate genes obtained from GWAS analysis. 

Round 2

Reviewer 2 Report

Author revised the article as per comments hence may be accepted for publication

Author Response

Dear reviewer, 

Thank you for your review and comments.